# PEneo: Unifying Line Extraction, Line Grouping, and Entity Linking for End-to-end Document Pair Extraction

Zening Lin
South China University of Technology
Guangzhou, China
eeznlin@mail.scut.edu.cn

Jiapeng Wang
South China University of Technology
Guangzhou, China
eejpwang@mail.scut.edu.cn

Teng Li
South China University of Technology
Guangzhou, China
ee_tengli@mail.scut.edu.cn

Wenhui Liao
South China University of Technology
Guangzhou, China
eelwh@mail.scut.edu.cn

Dayi Huang
Kingsoft Office
Zhuhai, China
huangdayi@wps.cn

Longfei Xiong
Kingsoft Office
Zhuhai, China
xionglongfei@wps.cn

Lianwen Jin*
South China University of Technology
Guangzhou, China
eelwjin@scut.edu.cn

## Abstract

Document pair extraction aims to identify key and value entities as well as their relationships from visually-rich documents. Most existing methods divide it into two separate tasks: semantic entity recognition (SER) and relation extraction (RE). However, simply concatenating SER and RE serially can lead to severe error propagation, and it fails to handle cases like multi-line entities in real scenarios. To address these issues, this paper introduces a novel framework, **PEneo** (**P**air **E**xtraction **n**ew **d**ecoder **o**ption), which performs document pair extraction in a unified pipeline, incorporating three concurrent sub-tasks: line extraction, line grouping, and entity linking. This approach alleviates the error accumulation problem and can handle the case of multi-line entities. Furthermore, to better evaluate the model's performance and to facilitate future research on pair extraction, we introduce RFUND, a re-annotated version of the commonly used FUNSD and XFUND datasets, to make them more accurate and cover realistic situations. Experiments on various benchmarks demonstrate PEneo's superiority over previous pipelines, boosting the performance by a large margin (e.g., 19.89%-22.91% F1 score on RFUND-EN) when combined with various backbones like LiLT and LayoutLMv3, showing its effectiveness and generality. Codes and the new annotations are available at https://github.com/ZeningLin/PEneo.

## CCS Concepts

• **Applied computing** → **Document analysis**; • **Computing methodologies** → **Computer vision**.

---

*Corresponding author.

*MM '24, October 28-November 1, 2024, Melbourne, VIC, Australia*
© 2024 Copyright held by the owner/author(s).
ACM ISBN 979-8-4007-0686-8/24/10
https://doi.org/10.1145/3664647.3680931

## Keywords

Visual Information Extraction, Document Analysis and Understanding, Vision and Language

**ACM Reference Format:**
Zening Lin, Jiapeng Wang, Teng Li, Wenhui Liao, Dayi Huang, Longfei Xiong, and Lianwen Jin. 2024. PEneo: Unifying Line Extraction, Line Grouping, and Entity Linking for End-to-end Document Pair Extraction. In *Proceedings of the 32nd ACM International Conference on Multimedia (MM '24), October 28-November 1, 2024, Melbourne, VIC, Australia.* ACM, New York, NY, USA, 10 pages. https://doi.org/10.1145/3664647.3680931

## 1 Introduction

Document pair extraction is a vital step in analyzing form-like documents containing information organized as key-value pairs. It involves identifying the key and value entities, as well as their linking relationships from document images. Previous research has generally divided it into two document understanding tasks: semantic entity recognition (SER) and relation extraction (RE). The SER task involves extracting contents that belong to predefined categories, such as retrieving store names and prices from receipts [13] or analyzing nutrition facts labels [16]. Most of the existing methods [12, 18, 29, 33, 34, 36] implement SER using BIO tagging, where tokens in the input text sequence are tagged as the beginning (B), inside (I), or outside (O) element for each entity. On the other hand, the RE task aims to identify relations between given entities, such as predicting the linkings between form elements [9, 35]. Previous works [29, 34, 36] have typically employed a linking classification network for relation extraction: given the entities in the document, it first generates representations for all possible entity pairs, then applies binary classification to filter out the valid ones. Document pair extraction is usually achieved by serially concatenating the above two tasks (SER+RE), where the SER model first identifies all the key and value entities from the document, and the RE model finds the matching values for each key. Figure 1 provides examples of SER, RE, and document pair extraction.

Although achievements have been made in SER and RE, the existing SER+RE approach overlooks several issues. In previous settings,

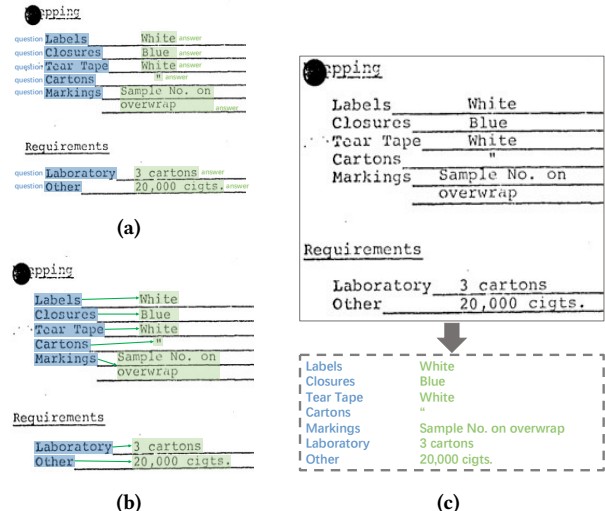

**Figure 1: Examples of SER, RE, and document pair extraction in [9]. (a) SER task, which aims at classifying fields into specific given entity categories. (b) RE task, which predicts the relations (green arrows) between the given entities. (c) Document pair extraction task that requires extraction of all key-value pairs from the document image.**

SER and RE are viewed as two distinct tasks that have inconsistent input/output forms and employ simplified evaluation metrics. For the SER task, entity-level OCR results are usually given [9], where text lines belonging to the same entity are aggregated and serialized in human reading order. The model categorizes each token based on the well-organized sequence, neglecting the impact of improper OCR outputs. In the RE task, models take the ground truths of the SER task as input, using prior knowledge of entity content and category. The model simply needs to predict the linkings based on the provided key and value entities, and the linking-level F1 score is taken as the evaluation metric. In real-world applications, however, the situation is considerably more complex. Commonly used OCR engines typically generate results at the line level. For entities with multiple lines, an extra line grouping step is required before BIO tagging, which is hard to realize for complex layout documents. Additionally, errors in SER predictions can significantly impact the RE step, resulting in unsatisfactory pair extraction results. Section 5.5 analyzes the SER+RE performance drop in detail.

To tackle the aforementioned challenges, we propose **PEneo** (**P**air **E**xtraction **n**ew **d**ecoder **o**ption) to implement pair extraction in a joint manner. Our framework begins by acquiring multi-modal representations of each token using an existing document understanding backbone. Then, a newly designed decoder concurrently performs the following three sub-tasks: (1) line extraction, which identifies the text lines belonging to the key and value entities; (2) line grouping, where lines within an entity are merged; (3) entity linking, which establishes the connections between keys and their corresponding values. The three tasks are optimized jointly to minimize their discrepancies and reduce error accumulation. Subsequently, a linking parsing module integrates the output from each

sub-task to generate the key-value pairs. This approach effectively suppresses errors in local predictions and produces optimal results. Notably, the decoder can collaborate with any BERT-like document understanding backbone and can be fine-tuned to downstream datasets directly without additional task-specific pre-training.

Furthermore, we found that some annotations in the two commonly used form understanding datasets, FUNSD [9] and XFUND [35], do not meet the real-world requirements well. Hence we propose their relabeled version, RFUND. The inconsistent labeling granularity in the original annotations is unified into line-level, aiming to imitate the output of a real OCR engine. We also rectified the category and relation annotations to make it more clear.

Our main contributions can be summarized as follows:

- We propose PEneo, a novel framework that unifies document pair extraction through joint modeling of line extraction, line grouping, and entity linking. The model has an enhanced error suppression capability and is able to cope with challenges like multi-line entities.
- We relabel the widely used FUNSD and XFUND datasets to better simulate real-world conditions for document pair extraction, including line-level OCR and more accurate annotations. The relabeled dataset is termed RFUND.
- Experiments on various benchmarks show that PEneo significantly outperforms existing pipelines when collaborating with different backbones, demonstrating the effectiveness and versatility of the proposed method.

## 2 Related Work

### 2.1 Document Pair Extraction Methods

Early studies [25, 31] utilized heuristic rules to extract key-value pairs from documents. These approaches exhibit limited applicability to specific document layouts and demonstrate poor generalization performance. In recent years, with the advancements in deep learning techniques, researchers have proposed several deep learning-based methods for pair extraction. LayoutLM [33] has first proposed embedding the coordinate of each word into a BERT-style model to capture the multi-modal features of each token. It also strengthens the model's representational capability with specially designed pre-training tasks. Subsequent works primarily focus on improving the backbones to obtain more powerful and general token representations. [10] introduces a novel relative spatial encoding to capture layout information effectively. [2, 7, 12, 17, 18, 36] incorporate visual features and enhance the interaction of different modalities through newly designed architectures and pre-training tasks. [29] proposes a two-branch structure that allows flexible switching of semantic encoding modules and fast adaptation in different language scenarios. To handle the text serialization problem, ERNIE-Layout [23] employs a Layout-Parser as well as a reading order prediction task to recognize and sort the text segments. To boost performance on RE, GeoLayoutLM [21] proposes a novel relation head and a geometric pre-training schema, and obtains outstanding performance on various RE benchmarks [22, 35]. ESP [37] introduces an end-to-end pipeline incorporating text detection, text recognition, entity extraction, and entity linking. It also predicts inter/intra-linkings between words to cope with multi-line

entities. The model shows outstanding performance on various relation extraction tasks. However, all of the aforementioned methods achieve pair extraction by simply concatenating the downstream SER and RE model, thereby overlooking the error accumulations in this process.

In addition to the SER+RE pipeline, some other approaches explore alternative ways to accomplish pair extraction. FUDGE [6] employs a graph-based detection scheme that iteratively aggregates text lines and predicts the key-value linking. Although it is capable of handling multi-line entities, its performance is relatively limited due to the absence of semantic information. SPADE [14] handles the document parsing task using the dependency parsing strategy, and it demonstrates good performance on the CORD [22] dataset. However, it models and decodes the document based on quantities of word-to-word relation, resulting in a huge computational overhead. Donut [15] and Dessurt [5] both propose image-to-sequence pipelines, which achieve pair extraction in a question-answering manner; QGN [3] introduces a query-driven network that generates the pair predictions using the value prefix. These generative models require a lot of training data and fail to handle complex layout documents. DocTr [19] first identifies anchor words from the input OCR results, then predicts entity-level bounding boxes and relations using a vision-language decoder, achieving structured information extraction and multi-line entity grouping. However, it requires task-specific pre-training and cannot be directly applied to existing backbones. TPP [39] employs a unified token path prediction framework for multi-line SER and RE tasks, and it outperforms conventional BIO-tagging baselines. However, TPP regards RE as a token clustering task, which leads to the inability to differentiate the key and value content individually. KVPFormer [11] takes a different approach by first identifying key entities in the given documents and then employing an answer prediction module to determine their corresponding values. Notably, its proposed spatial compatibility feature helps achieve exceptional performance in the RE task without pre-training, but it requires prior knowledge of entity spans and cannot handle unordered OCR inputs.

## 2.2 Joint Extraction in Plain Texts

Joint extraction aims to identify the subject-relation-object triplets simultaneously from plain texts. Mainstream approaches can be broadly categorized into two types: pipeline-based methods and joint methods.

The pipeline-based method is akin to the SER+RE approach mentioned above. It involves a sequential combination of SER and RE tasks, wherein the entity contents are initially predicted, followed by the classification of the semantic relation type between them [27, 28, 38, 41]. The joint methods unify the two tasks through specifically designed network architecture. [4, 40] propose novel tagging schemes that incorporate relation extraction annotations into BIO tags, thereby unifying the entire pipeline in BIO tagging manners. [32] employs span prediction to extract the subject content from the token sequence, subsequently predicting their corresponding objects and relation types through relation-specific object taggers. [30] leverages span prediction for entity identification, then constructs head and tail linking matrices for relation extraction. This approach enables the parsing of triplets through a joint decoding schema.

It is noteworthy that while there are valuable insights to be gained from joint extraction in plain texts, the extraction of key-value pairs in visually-rich documents presents unique challenges that necessitate additional efforts. For instance, the entities to be extracted may span across multiple OCR boxes, giving rise to challenges related to multi-line SER. Moreover, the determination of relationships between entities relies on spatial information, thus requiring the development of specialized modules to effectively address these requirements.

## 3 The RFUND Dataset

### 3.1 Analysis of the Original Annotations

The FUNSD dataset is a commonly used form understanding benchmark that comprises scanned English documents. The XFUND dataset is its multilingual extension, covering 7 languages (Chinese, Japanese, Spanish, French, Italian, German, and Portuguese). Entities in these forms are categorized into four types, including *header*, *question*, *answer*, and *other*. Entity-level and word-level OCR results are provided, and linking relationships between different entities are annotated to represent the structure of the form.

While most contents in FUNSD are annotated at the entity level, multi-line entities with first-line indentation are annotated in a distinct manner. As illustrated in Figure 2a, the first line of the answer paragraph is considered a separate entity, while the other lines remain aggregated as another. The question entity is linked to both answers, leading to redundant annotations. XFUND exhibits variable granularity in annotations, with some contents labeled at the entity level and others at the line level, as shown in Figure 2b. Such inconsistent labeling standards can hinder model training and fail to work in real-world scenarios. Moreover, it is observed that certain entities in both FUNSD and XFUND have category labels that differ from human understanding (Figure 2c), highlighting the need for refinement of the annotations in this aspect.

### 3.2 Relabeling for Real-World Scenarios

Based on the original entity-level and word-level OCR results, we implemented a set of rules to divide paragraphs into line-level text strings and bounding boxes. Specifically, we examined the vertical distance between two adjacent words within an entity. If the difference exceeds the average height of entity words, we assign the latter word to the subsequent line. For cases that could not be handled perfectly by the rules, we made manual corrections. To represent entity-level information, line grouping annotations were added to indicate the correct order of line aggregation. Cases of first-line indent described above were corrected, and any redundant linking labels were removed. To ensure consistency with human understanding, we adjusted the entity category labels and key-value linkings accordingly. In addition, we eliminated header-to-question linkings that describe nested information, simplifying the task scope. We term the resulting dataset as RFUND, and Table 1 summarizes its key statistics.

### 3.3 Task Definition

The model is expected to take the line-level OCR results, which are commonly used in real scenarios, as input. It should then predict

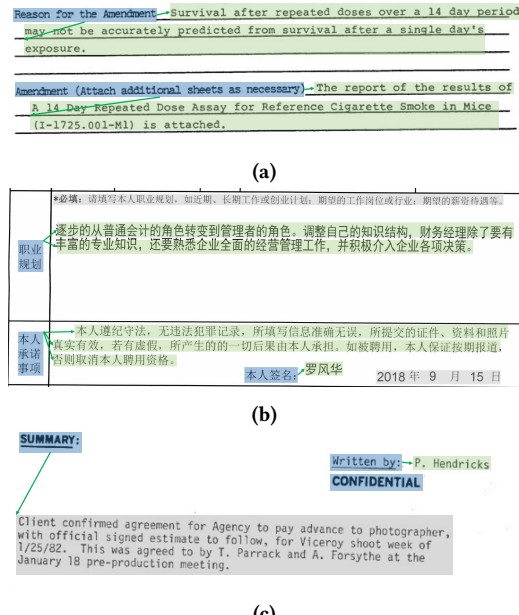

(a)

(b)

(c)

**Figure 2: Examples of the original FUNSD and XFUND annotations. Boxes in blue, green, and grey stand for *question*, *answer*, and *other* entities, respectively. Green arrows refer to key-value linkings. (a) Annotations for entities with first-line indentation in FUNSD. (b) Inconsistent labeling granularity in XFUND, keys are labeled at entity level, while values are at line level. (c) Confusing annotations, *answer* entity "Client confirmed agreement ..." was labeled as *other*, while the *other* entity "CONFIDENTIAL" was labeled as the *question*.**

**Table 1: Key statistics of the RFUND dataset**

| Lang | Split | # of entities | # of multi-line entities | # of pairs |
|------|-------|---------------|--------------------------|------------|
| EN | **train** | 7049 | 631 | 3023 |
|    | **test** | 2201 | 277 | 848 |
| ZH | **train** | 9948 | 1139 | 3887 |
|    | **test** | 3469 | 435 | 1414 |
| JA | **train** | 9775 | 778 | 2875 |
|    | **test** | 3390 | 342 | 1094 |
| ES | **train** | 11109 | 521 | 4022 |
|    | **test** | 3354 | 180 | 1186 |
| FR | **train** | 8680 | 307 | 3444 |
|    | **test** | 3499 | 153 | 1404 |
| IT | **train** | 11720 | 581 | 5111 |
|    | **test** | 3769 | 207 | 1635 |
| DE | **train** | 8177 | 575 | 3500 |
|    | **test** | 2645 | 202 | 1086 |
| PT | **train** | 11259 | 591 | 4211 |
|    | **test** | 4101 | 179 | 1593 |

all the key-value pairs in string format. Pair-level F1 score is employed for performance comparison, where a pair prediction will be considered as True Positive if and only if both the predicted key and value text strings exactly match the ground truths.

## 4  PEneo

The architecture of our proposed framework is depicted in Figure 3. PEneo provides a new decoder for the pair extraction task. It first derives token representations using existing multi-modal document understanding backbones like LayoutLMv2. The decoder then concurrently generates relation matrices for three sub-tasks—line extraction, line grouping, and entity linking. Finally, a linking parsing algorithm is applied to obtain the predicted key-value pairs. We elaborate on each module in the following sections.

### 4.1  Multi-modal Encoder

The encoder tokenizes the input text lines into tokens and integrates semantic, layout, and visual (optional) information to obtain multi-modal features for each token. Various BERT-like document understanding models, including LayoutLMv2 [36], LayoutLMv3 [12], and LiLT [29], can serve as the encoding backbone for PEneo. To reduce memory consumption for the following operation, we append a linear projection layer to map the channels of the output features to a smaller size at the final stage:

$$\mathbf{h}_i = \mathbf{W}_{proj}\mathbf{f}_i + \mathbf{b}_{proj}, \tag{1}$$

where $\mathbf{f}_i \in \mathbb{R}^{c_e}$ is the backbone output feature, $\mathbf{h}_i \in \mathbb{R}^{c_d}$ is the channel reduced feature, $\mathbf{W}_{proj} \in \mathbb{R}^{c_d \times c_e}$ and $\mathbf{b}_{proj} \in \mathbb{R}^{c_d}$ are parameters of the projection layer. $c_e$ is the output size of the backbone, and $c_d$ is the reduced channel size.

### 4.2  Joint Extraction Decoder

The decoder is responsible for both entity recognition and linking prediction. Specifically, it performs the following operations: (1) **Line Extraction**: Identifies the text lines belonging to the key and value entities. (2) **Line Grouping**: Merges lines within an entity to create cohesive representations. (3) **Entity Linking**: Establishes connections between key and value entities. These operations are optimized jointly to minimize discrepancies and reduce error accumulation, ensuring the overall effectiveness of PEneo.

Inspired by [30], token representations $\mathbf{h}_i$ from the encoder are concatenated in a pair-wise manner. Subsequently, a pair encoding layer is applied to obtain the token pair representations matrix $\mathbf{M} \in \mathbb{R}^{N \times N \times c_d}$, where $N$ is the number of input tokens. Each entry $\mathbf{M}_{ij}$ is computed as

$$\mathbf{M}_{ij} = \mathbf{W}_{pair}(\mathbf{h}_i \oplus \mathbf{h}_j) + \mathbf{b}_{pair}. \tag{2}$$

Here $\oplus$ denotes vector concatenation. $\mathbf{W}_{pair} \in \mathbb{R}^{c_d \times 2c_d}$ and $\mathbf{b}_{pair} \in \mathbb{R}^{c_d}$ are parameters of the pair encoding layer.

The matrix $M$ is then fed into three separate branches, performing line extraction, line grouping, and entity linking tasks in parallel.

*4.2.1  Line Extraction.* This branch extracts lines belonging to key and value entities through span prediction. A classifier is applied to $\mathbf{M}$ to get the line extraction score $\mathbf{P}^{(le)} \in \mathbb{R}^{N \times N \times 2}$:

$$\mathbf{P}^{(le)} = softmax(MLP^{(le)}(\mathbf{M})). \tag{3}$$

The prediction matrix $\mathbf{M}^{(le)} \in \mathbb{R}^{N \times N}$ is obtained through argmax operation on $\mathbf{P}^{(le)}$, identifying the start and end tokens of lines

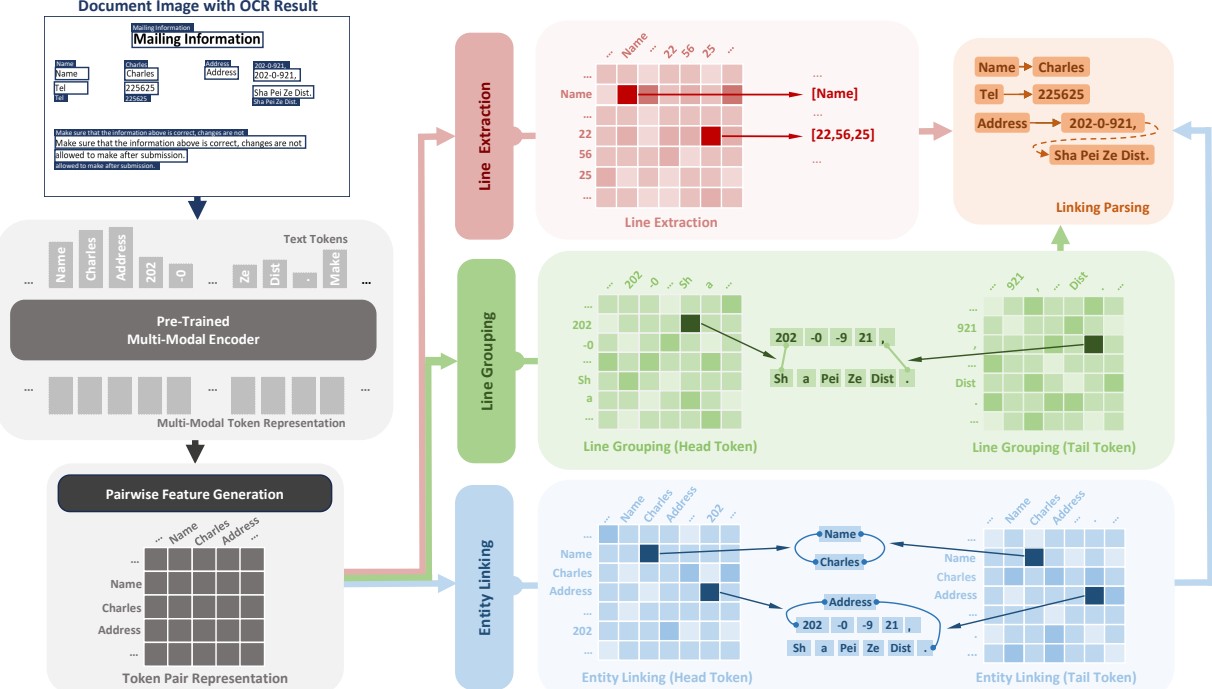

**Figure 3: Model architecture of PEneo. Line-level OCR results are processed by the Pre-Trained Multi-Modal encoder to get representations of each token. The decoder then generates pair-wise features and applies Line Extraction, Line Grouping, and Entity Linking to obtain predictions of line spans, line aggregation, and key-value linkings. Finally, the Line Parsing Module integrates the predictions above to generate key-value pairs.**

that pertain to key or value entities, in which entries are defined as

$$\mathbf{M}_{ij}^{(le)} = \begin{cases} 1, & \text{tokens in (i, j) form a key/value line} \\ 0, & \text{otherwise.} \end{cases} \quad (4)$$

As shown in Figure 3, the element at row *Name*, column *Name* indicates that a text line consists of a single token *Name* is extracted. While element at row *22*, column *25* indicates that tokens *22*, *56*, and *25* form a target line *225625*.

*4.2.2 Line Grouping.* To aggregate lines belonging to the same entity, we create a line head grouping matrix $\mathbf{M}^{(lgh)}$ and a line tail grouping matrix $\mathbf{M}^{(lgt)}$ to represent the connections between the tokens at the beginning and end of each line, respectively. For two neighboring lines within an entity, whose tokens range from $(a, b)$ and $(c, d)$, their connections are represented by $\mathbf{M}_{ac}^{(lgh)} = 1$ and $\mathbf{M}_{bd}^{(lgt)} = 1$. In Figure 3, entity *202-0-921, Sha Pei Ze Dist.* consists of two text lines *202-0-921*, and *Sha Pei Ze Dist.*. Linking predictions between their head tokens *202* and *Sh*, as well as their tail tokens *,* and *.* indicate that these two lines should be grouped.

*4.2.3 Entity Linking.* This branch predicts linkings between key and value entities. Two classifiers are applied to $\mathbf{M}$, forming the entity head linking matrix $\mathbf{M}^{(elh)}$ and the entity tail linking matrix $\mathbf{M}^{(elt)}$. For a key-value pair where the key ranges from tokens

$(e, f)$ and the value ranges from tokens $(g, h)$, we have $\mathbf{M}_{eg}^{(elh)} = 1$ and $\mathbf{M}_{fh}^{(elt)} = 1$. If the key/value entity contains multiple text lines, we establish a connection from the head token of the key entity's first line to the head token of the value entity's first line, as well as a connection from the tail token of the key entity's last line to the tail token of the value entity's last line. As shown in Figure 3, the key entity *Address* should be connected to value *202-0-921, Sha Pei Ze Dist.*. Hence the connections between the head tokens of their first lines (*Address* and *202*) and the tail tokens of their last lines (*Address* and .) are predicted as positive.

*4.2.4 Linking Parsing.* Matrix $\mathbf{M}^{(elh)}$ predicts the first token of each key and value entity. The Span of the entity's first line can be determined by referring to the line extraction result from $\mathbf{M}^{(le)}$. For multi-line entities, based on the first and last token of each line, spans of the entity's following lines can be retrieved iteratively from the line grouping predictions $\mathbf{M}^{(lgh)}$ and $\mathbf{M}^{(lgt)}$. Once we collect all the contents of the current pair, we compare the last token of the key and value entity with the entity tail linking result $\mathbf{M}^{(elt)}$ to determine the validity of the current prediction. During the parsing process, if the predictions of different matrices are found to be contradictory, the current parsed content is considered to be erroneous and is directly eliminated.

## 4.3 Supervised Learning Target

For each prediction matrix, we adopt a weighted cross-entropy loss as its supervised learning target:

$$\mathcal{L}_* = CrossEntropy\left(\mathbf{P}^{(*)}, \mathbf{Y}^{(*)}; \mathbf{w}\right), \qquad (5)$$

where $\mathbf{P}^{(*)}$ is the prediction score matrix of each branch, $\mathbf{Y}^{(*)}$ is the corresponding label. $\mathbf{w}$ is the class weighting tensor.

The overall loss of PEneo during the training phase is the weighted sum of losses from the five matrices.

$$\mathcal{L} = \lambda_1 \mathcal{L}_{le} + \lambda_2 \mathcal{L}_{lgh} + \lambda_3 \mathcal{L}_{lgt}$$
$$+ \lambda_4 \mathcal{L}_{elh} + \lambda_5 \mathcal{L}_{elt}, \qquad (6)$$

where $\mathcal{L}_{le}$ stands for the line extraction loss, $\mathcal{L}_{lgh}$ and $\mathcal{L}_{lgt}$ stand for the line head and tail grouping losses, $\mathcal{L}_{elh}$ and $\mathcal{L}_{elt}$ stand for the entity head and tail linking losses, $\lambda_i, i = 1, 2, \cdots, 5$ are loss weighting hyper-parameters.

## 5 Experiments

## 5.1 Datasets

We conduct experiments on RFUND and SIBR [37]. As described in section 3, **RFUND** contains 8 subsets corresponding to 8 different languages. We follow the language-specific fine-tuning settings in [34] to evaluate the model's performance on each subset. **SIBR** is a bilingual dataset composed of 600 training samples and 400 testing samples. It contains 600 Chinese invoices, 300 English bills of entry, and 100 bilingual receipts. The dataset is annotated at line level, with entity linking (inter-links) and line grouping (intra-links) labels provided. We observed some contradictory annotations in SIBR and made manual corrections. In our experiment, we focus on the linkings between question and answer entities in SIBR and employ pair-level F1 score as the evaluation metric.

## 5.2 Implementation Details

The reduced feature channel size $c_d$ is set to $c_e/2$. During the training phase, the loss weighting hyper-parameters $\lambda_i, i = 1, 2, \cdots, 5$ are all set to 1. To address category imbalance, the class weighting tensor for the cross-entropy loss is set to $[1, 10]$. We employ AdamW [20] as the optimizer. The learning rate is set to 2e-6 for the encoder backbone and 1e-4 for the decoder, scheduled by a linear scheduler with a warm-up ratio of 0.1. We fine-tune PEneo for 650 epochs on RFUND and 330 epochs on SIBR, with a batch size of 4.

## 5.3 Baseline Settings

We employed several widely used and publicly available models, LiLT, LayoutLMv2, LayoutXLM, and LayoutLMv3, as the encoding backbone to evaluate the effectiveness of our proposed PEneo framework. The baseline method serially combines an SER model and a RE model for pair extraction. At the SER stage, we sorted the input lines with Augmented XY Cut [8], aiming to maximize the adjacency of lines within an entity. The model learns to extract entities based on the sorted sequence and group the correctly ordered lines through BIO tagging[24]. For the RE part, we train the model using entity-level annotations, consistent with previous studies. Additionally, we adapted FUDGE [6], Donut [15], GeoLayoutLM [21], TPP [39] and GPT-4V [1] to the pair extraction task. FUDGE

**Table 2: Comparison with existing methods on pair extraction with RFUND-EN. † means that the RE module is re-implemented by us. ‡ means that the metric has been adjusted to be less stringent as a compromise.**

| Method | Venue | Pipeline | F1 |
|---|---|---|---|
| FUDGE [6] | ICDAR'21 | End-to-End | 53.15‡ |
| Donut [15] | ECCV'22 | Image2Seq | 24.54 |
| GeoLayoutLM [21] | CVPR'23 | SER+RE | 69.03 |
| TPP-LayoutLMv3$_{BASE}$ [39] | EMNLP'23 | Joint | 50.27‡ |
| GPT-4V w/o OCR [1] | arXiv'23 | Image2Seq | 20.96 |
| GPT-4V w OCR [1] | arXiv'23 | Image2Seq | 38.15 |
| LiLT[EN-R]$_{BASE}$ [29] | ACL'22 | SER+RE | 54.33 |
| **PEneo-LiLT[EN-R]$_{BASE}$** | Ours | Joint | 74.22 (**+19.89**) |
| LiLT[InfoXLM]$_{BASE}$ [29] | ACL'22 | SER+RE | 52.18 |
| **PEneo-LiLT[InfoXLM]$_{BASE}$** | Ours | Joint | 74.29 (**+22.11**) |
| LayoutXLM$_{BASE}$ [34] | arXiv'21 | SER+RE | 52.98 |
| **PEneo-LayoutXLM$_{BASE}$** | Ours | Joint | 74.25 (**+21.27**) |
| LayoutLMv2$_{BASE}$ [36] | ACL'21 | SER+RE | 49.06 |
| **PEneo-LayoutLMv2$_{BASE}$** | Ours | Joint | 71.97 (**+22.91**) |
| LayoutLMv3$_{BASE}$ [12] | ACMMM'22 | SER+RE | 57.66† |
| **PEneo-LayoutLMv3$_{BASE}$** | Ours | Joint | 79.27 (**+21.61**) |

**Table 3: Performance comparison on SIBR dataset. † means that the RE module is re-implemented by us.**

| Method | Venue | Pipeline | F1 |
|---|---|---|---|
| Donut$_{BASE}$ [29] | ECCV'22 | Image2Seq | 17.26 |
| LiLT[InfoXLM]$_{BASE}$ [29] | ACL'22 | SER+RE | 72.76 |
| **PEneo-LiLT[InfoXLM]$_{BASE}$** | Ours | Joint | 82.36 (**+9.60**) |
| LayoutXLM$_{BASE}$ [34] | arXiv'21 | SER+RE | 70.45 |
| **PEneo-LayoutXLM$_{BASE}$** | Ours | Joint | 82.23 (**+11.78**) |
| LayoutLMv3$_{Chinese\ BASE}$ [12] | ACMMM'22 | SER+RE | 73.51† |
| **PEneo-LayoutLMv3$_{Chinese\ BASE}$** | Ours | Joint | 82.52 (**+9.01**) |

employs a graph-based detection pipeline that predicts the key-value bounding box pairs. Since it only predicts boxes, we report its performance on the box-pair F1-score as a compromise. Donut takes the document image as input and predicts HTML-like strings containing key-value pairs. GeoLayoutLM is a strong baseline that includes a powerful RE decoder, and we perform pair extraction by concatenating its downstream SER and RE models, following the aforementioned SER+RE settings. TPP employs a token clustering scheme, and the groups of pair tokens can be obtained by parsing its VrD-EL matrix with depth-first searching. Its performance is reported on token-group F1-score. For GPT-4V, we follow the evaluation pipeline proposed by [26]. It is worth noting that some models only provide pre-trained weights for a single or a small number of languages, which does not cover all the samples in RFUND. Hence, we only evaluated the model's performance on the language subset covered by their pre-training corpus.

## 5.4 Comparison with Existing Methods

Results are shown in Table 2-4. Previous pipelines underperform on the pair extraction task. FUDGE and Donut utilize the visual modality only, which may be insufficient for analyzing the diverse and

**Table 4: Performance comparison on RFUND's multilingual subsets. - means that the model does not provide pre-trained weights that cover the corresponding language. † means that the RE module is re-implemented by us. Results are reported in F1-score.**

| Method | ZH | JA | ES | FR | IT | DE | PT |
|---|---|---|---|---|---|---|---|
| Donut$_{BASE}$ [15] | 28.21 | 13.82 | - | - | - | - | - |
| LayoutLMv3$_{Chinese\ BASE}$ [12] | 72.14$^{†}$ | - | - | - | - | - | - |
| **PEneo-LayoutLMv3$_{Chinese\ BASE}$** | 85.05 (+12.91) | - | - | - | - | - | - |
| LiLT[InfoXLM]$_{BASE}$ [29] | 66.50 | 43.98 | 63.85 | 62.60 | 60.57 | 55.13 | 52.96 |
| **PEneo-LiLT[InfoXLM]$_{BASE}$** | 80.51 (+14.01) | 54.59 (+10.61) | 71.43 (+7.58) | 77.49 (+14.89) | 73.62 (+13.05) | 70.11 (+14.98) | 71.43 (+18.47) |
| LayoutXLM$_{BASE}$ [34] | 64.11 | 40.21 | 66.75 | 67.98 | 63.04 | 58.77 | 59.79 |
| **PEneo-LayoutXLM$_{BASE}$** | 80.41 (+16.30) | 52.81 (+12.60) | 74.56 (+7.81) | 78.11 (+10.13) | 75.17 (+12.13) | 74.06 (+15.29) | 70.81 (+11.02) |

complex key-value relationships in the document. We argue that Donut performs better in scenarios with a small number of output tokens, such as receipt understanding in CORD [22]. Its prediction ability is not yet fully developed for documents with a large number of texts. TPP employs a unified token-path-prediction pipeline to tackle the RE task, without a dedicated design for error suppression. In this schema, even a minor error in the linking prediction matrix could result in a completely incorrect outcome. For GPT-4V, integrating the OCR results into the prompt can significantly improve its performance, but it still falls short of numerous supervised approaches. Upon analyzing its output, we argue that its underperformance can be attributed primarily to the LLM hallucination, which leads to numerous redundant or inaccurate predictions. The SER+RE pipelines suffer from performance drop, mainly due to the error accumulation between modules and the improper text order, which will be discussed in the subsequent sections. PEneo, on the other hand, substantially improves the performance of each backbone. On RFUND-EN, the F1 score of the entire pipeline is boost by 19.89% for LiLT[EN-R]$_{BASE}$, 22.11% for LiLT[InfoXLM]$_{BASE}$, 21.27% for LayoutXLM$_{BASE}$, 22.91% for LayoutLMv2$_{BASE}$, and 21.61% for LayoutLMv3$_{BASE}$. Most of these backbones outperform the strong baseline GeoLayoutLM which contains task-specific pre-trained RE modules, although there exist huge gaps between them under the previous SER+RE setting. For the other language subset of RFUND, PEneo still offers substantial performance improvements. On the SIBR dataset, the new pipeline has demonstrated a score improvement ranging from 9.01% to 11.78%, confirming its ability in bilingual settings. These outcomes underscore the effectiveness and versatility of PEneo, as it consistently achieves performance gains across multiple language scenarios and diverse backbone configurations.

## 5.5 Analysis of Module Collaboration

*Performance Drop in the SER+RE Pipeline.* For the SER+RE pipeline, although the downstream models may work well on the SER or RE task, the performance drops drastically when it comes to the pair extraction setting. Figure 4 visualizes the failure cases. Erroneous predictions in the SER step greatly confuse the RE module, leading to redundant or missing output. The imperfect line ordering generated by the preprocessing step also makes it difficult to group entity lines through BIO tagging.

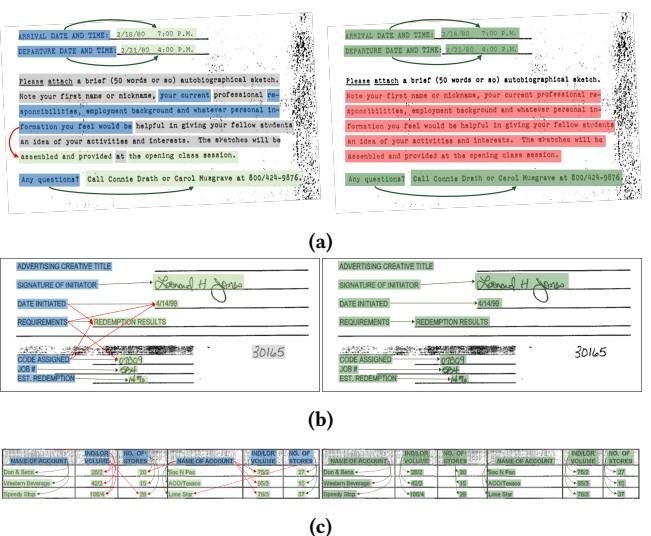

(a)

(b)

(c)

**Figure 4: Performance comparison between PEneo and SER+RE. Left: prediction of SER+RE. Blue, green, and grey boxes indicate prediction for question, answer, and other entities, respectively. Right: prediction of PEneo. The green boxes are correctly extracted lines or entities, red are false positives. The green arrows are correct pair predictions, and the red arrows are wrong.**

We observe that LiLT and LayoutLMv3 underperform in several cases, which seems to be contradictory to the results reported in previous literature [12, 29] at first glance. In fact, these two methods utilize entity-level boxes for layout modeling, while the conventional settings [21, 33, 34, 36] use word-level boxes. In our experiment, all the models take the line-level coordinates as input, which affects their SER ability to some extent.

To further illustrate the performance drop phenomenon in the SER+RE pipeline, we conducted experiments on RFUND-EN with LiLT[InfoXLM]$_{BASE}$ using different SER results. As shown in Table 5, a performance gap of 19 points in the SER module results in a 15-point decrease in pair extraction performance (#1 and #2). As the SER score is further reduced by 3 points, the performance of the pair extraction reduces by 3 points as well (#2 and # 3, #4). Results show that the SER+RE approach exhibits a serious accumulation of errors.

**Table 5: Error accumulation in the SER+RE pipeline. Results are reported in F1 score.**

| # | SER | RE | Pair Extraction |
|---|-----|-----|-----------------|
| 1 | 100.00 | 67.18 | 67.18 |
| 2 | 80.28 (-19.72) | 67.18 | 52.18 (-15.00) |
| 3 | 79.20 (-20.80) | 67.18 | 51.09 (-16.09) |
| 4 | 76.88 (-23.12) | 67.18 | 48.42 (-18.76) |

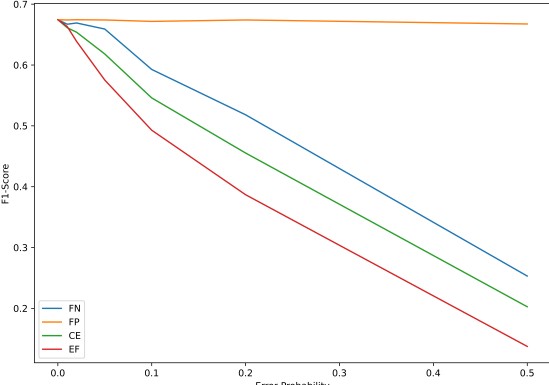

**Figure 5: Impact of different SER results on pair extraction performance. FN refers to entity false negative, FP refers to entity false positive, CE refers to entity category error, and EF refers to entity fragmentation.**

The accuracy of the SER module greatly affects the performance of the whole pipeline.

We then explore the impact of different types of SER errors on the whole pipeline. Commonly seen errors in the SER step include: (1) entity false negative, where keys and values are categorized as background elements; (2) entity false positive, where background elements are categorized as keys or values; (3) entity category error, where keys/values are identified as values/keys; (4) entity fragmentation, where an entity is recognized as multiple parts belong to different categories, as shown in Figure 4a. We added the above disturbances to the SER ground truths with different probabilities and tested the performance variation of pair extraction under the setting of a fixed RE module. Results are shown in Figure 5. Entity fragmentation has the largest effect since inaccurate entity spans inherently lead to mistakes. False positives have the smallest influence as the RE module could filter out some misclassifications given accurate spans. False negatives directly cause missing links, while category errors interfere with the RE module's reasoning. Overall, precise entity span detection proved critical for the SER+RE pipeline. However, it is highly dependent on the correct order of input text and large granularity of coordinate information [18], which is often difficult to achieve in practice, making the model underperform.

*Effectiveness of PEneo.* Compared with the SER+RE scheme, our approach suppresses the error accumulation between modules, reduces the influence brought by other factors to different components, and the capacity of the backbone is fully exploited. Figure 4 shows how PEneo suppresses the errors. In Figure 4a, although the

**Table 6: Performance analysis of PEneo using predicted outputs (first row) vs. ground truth (second, third, and fourth row) of line extraction and line grouping. LiLT-I refers to LiLT[InfoXLM]$_{\text{BASE}}$, and LaLM3B refers to LayoutLMv3$_{\text{BASE}}$. Results are reported in F1 score.**

| Encoder | Line Extraction | Line Grouping | Pair Extraction |
|---------|-----------------|---------------|-----------------|
| LiLT-I | 87.68 | 53.87 | 74.29 |
|  | 100.00 (+12.32) | 53.87 | 74.93 (+0.64) |
|  | 87.68 | 100.00 (+46.13) | 77.14 (+2.85) |
|  | 100.00 (+12.32) | 100.00 (+46.13) | 78.85 (+4.56) |
| LaLM3B | 92.84 | 63.44 | 79.27 |
|  | 100.00 (+7.16) | 63.44 | 80.18 (+0.91) |
|  | 92.84 | 100.00 (+36.56) | 82.25 (+2.98) |
|  | 100.00 (+7.16) | 100.00 (+36.56) | 83.44 (+4.17) |

line extraction module of PEneo initially raised some background elements, they were filtered out by referring to the line grouping and entity linking predictions. In Figure 4b, PEneo gives correct results through the cooperation of different modules, while the SER+RE method gives false positive predictions. In Figure 4c, the SER+RE pipeline fails to group the multi-line entity *IND/LOR VOL-UME* in the table header, leading to erroneous predictions. PEneo, on the other hand, successfully addresses this challenge.

To further illustrate the advantages of PEneo, we replace the predictions of line extraction and line grouping with ground truths to test for variations in pair extraction performance. As shown in Table 6, in contrast to the SER+RE pipeline, using true labels in line extraction and line grouping only led to minor pair extraction gains, despite huge gaps between the performance of the predictions and ground truths. This demonstrates PEneo's ability to suppress downstream errors, thanks to the joint modeling and evidence accumulation of the three sub-tasks. Incorrect local predictions can be effectively rectified during the final linking parsing stage.

## 6 Conclusion and Future Work

In this paper, we proposed PEneo, a novel framework for end-to-end document pair extraction from visually-rich documents. By unifying the line extraction, line grouping, and entity linking tasks into a joint pipeline, PEneo effectively addressed the error propagation and challenges associated with multi-line entities. Experiments show that the proposed method outperforms previous pipelines by a large margin when collaborating with various backbones, demonstrating its effectiveness and versatility. Additionally, we introduced RFUND, a re-annotated version of the widely used FUNSD and XFUND datasets, to provide a more accurate and practical evaluation in real-world scenarios. Future work will focus on improving robustness to imperfect OCR results and complex structure parsing to enhance applicability to real-world documents. Investigating techniques like multi-task learning across the sub-tasks could further improve joint modeling. Overall, we hope this work will spark further research beyond the realms of prevalent SER+RE pipelines, and we believe the proposed PEneo provides an important step towards unified, real-world document pair extraction.

## Acknowledgments

This research is supported in part by National Natural Science Foundation of China (Grant No.: 62441604, 61936003).

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
