# OpenReview forum: "PEneo: Unifying Line Extraction, Line Grouping, and Entity Linking for End-to-end Document Pair Extraction"
_acmmm.org/ACMMM/2024/Conference — MM2024 Poster_

### Official Review · Reviewer_NaLy · 2024-05-19

**Rating:** 2
**Confidence:** 3

**Summary:**

This paper proposes PEneo, a novel framework that unifies line extraction, line grouping, and entity linking for end-to-end document pair extraction. The authors argue that the existing SER+RE pipeline approach suffers from error accumulation and is unable to handle multi-line entities. PEneo addresses these issues by concurrently performing the three sub-tasks and optimizing them jointly. The authors also introduce RFUND, a relabeled version of the FUNSD and XFUND datasets, to better simulate real-world scenarios. Experiments on various benchmarks demonstrate PEneo's superiority over previous pipelines when combined with different backbones.

**Strengths:**

1. The proposed PEneo framework provides a novel approach to unify line extraction, line grouping, and entity linking, which helps alleviate the error accumulation problem in the existing SER+RE pipeline.

2. PEneo is designed to handle multi-line entities, which is a common challenge in real-world scenarios.

3. The introduction of the RFUND dataset provides more accurate annotations and covers realistic situations, facilitating future research on pair extraction.

**Limitations:**

1. The paper only compares PEneo with the SER+RE pipeline and Donut, which is not a strong enough baseline. The authors should compare their method with more recent and advanced approaches, such as FUDGE [5], SPADE [13], DocTr [18], TPP [37], and KVPFormer [10], to demonstrate the effectiveness of their proposed framework.

2.  While the authors introduce the RFUND dataset and highlight its improvements over the original FUNSD and XFUND datasets, they do not provide a detailed analysis of how these changes impact the performance of various methods. A more thorough evaluation of the dataset's characteristics and its influence on different approaches would strengthen the paper's contributions.

3. The authors should consider adding visualizations or examples to better illustrate the differences between the original FUNSD and XFUND annotations and the relabeled RFUND dataset.

**Suitability:**

3

---

### Official Review · Reviewer_qAWH · 2024-05-25

**Rating:** 5
**Confidence:** 2

**Summary:**

This paper proposes PEneo, a novel unified framework for document pair extraction that jointly models three sub-tasks: line extraction, line grouping, and entity linking. By concurrently optimizing these sub-tasks using a newly designed decoder, PEneo alleviates error propagation and can effectively handle multi-line entities. The framework leverages existing document understanding backbones like LayoutLMv3 and LiLT to encode multimodal token representations, which are then processed by the proposed decoder to generate the key-value pairs. Additionally, the authors introduce RFUND, a re-annotated version of the FUNSD and XFUND datasets, to better simulate real-world scenarios and provide more accurate annotations. Extensive experiments demonstrate PEneo's significant performance gains over traditional pipelined approaches on various benchmarks.

**Strengths:**

1. Unifies the three sub-tasks (line extraction, line grouping, and entity linking) of document pair extraction through joint modeling, reducing error accumulation.
2. Capable of handling multi-line entities without requiring an additional line grouping step.
4. Significantly outperforms existing pipeline methods on multiple benchmarks.

**Limitations:**

Comparison with Recent Generative Models: It would be interesting to compare the performance of our model with the latest multimodal large language models, such as GPT-4 or Claude, on this task. These models may have the capability to perform zero-shot or few-shot learning on the task, potentially achieving better results without the need for extensive fine-tuning.

**Suitability:**

2

---

### Official Review · Reviewer_3d1i · 2024-05-25

**Rating:** 4
**Confidence:** 1

**Summary:**

This paper introduces PEneo (Pair Extraction new decoder option), which performs document pair extraction in a unified pipeline, incorporating three concurrent sub-tasks: line extraction, line grouping, and entity linking. This approach alleviates the error accumulation problem and can handle the case of multi-line entities.

**Strengths:**

The topic is interesting and related to the MM community.
The paper is well-written and easy to follow. The experiments are comprehensive and the results are promising.

**Limitations:**

Baselines are old. Most of the baselines are from 2021 to 2022. It would be better to compare with more recent methods, or consider multi-modal LLMs.
No scalability analysis or time complexity analysis is provided.

**Suitability:**

3

---

### Official Review · Reviewer_jHXc · 2024-05-30

**Rating:** 3
**Confidence:** 3

**Summary:**

The paper introduces PEneo, a novel framework for document pair extraction that addresses the shortcomings of existing methods by integrating three concurrent sub-tasks: line extraction, line grouping, and entity linking. This unified approach reduces error propagation and effectively handles multi-line entities. Additionally, the paper presents RFUND, a re-annotated version of the FUNSD and XFUND datasets.

**Strengths:**

- PEneo's approach to unifying line extraction, line grouping, and entity linking into a single pipeline is innovative and addresses the limitations of traditional two-step methods.
- The framework's compatibility with various document understanding backbones makes it versatile and adaptable to different document types and languages.
- PEneo significantly outperforms previous methods, achieving substantial performance gains across various benchmarks when used with different backbones like LiLT and LayoutLMv3.

**Limitations:**

- The experiments are not comprehensive enough, as only a few models were tested. The authors should compare PEneo with more models like TRIE+ and ESP, and report results on the original datasets to ensure fairness, in addition to RFUND.
- The construction of the re-annotated RFUND dataset is commendable, but the paper lacks detailed descriptions of RFUND. Information such as the error rate in the original datasets, the number of annotators involved, and whether cross-validation was conducted on the annotations is missing.
- The paper does not make full use of its available space, with nearly a page left incomplete. The authors could consider including content from the supplementary materials in the main text.

**Suitability:**

3

---

### Meta-Review · Area_Chair_bz12 · 2024-07-03

**Recommendation:** Accept (Poster)
**Confidence:** 4

**Metareview:**

The paper has notable strengths highlighted such as the innovative unification of line extraction, line grouping, and entity linking in the PEneo framework, which addresses the limitations of traditional methods. The framework's versatility with various document understanding backbones and its significant performance improvements across benchmarks are commended. However, several limitations were also noted, including the need for more comprehensive experiments with additional models, a lack of detailed descriptions and analysis of the re-annotated RFUND dataset, and insufficient comparison with recent advanced approaches and multimodal large language models. Additionally, reviewers suggested including content from supplementary materials and providing visualizations to enhance the paper's clarity and impact. Given these points, while the paper demonstrates strong potential and innovation, addressing the identified limitations could significantly strengthen its contributions.